# Fascia Layer—A Novel Target for the Application of Biomaterials in Skin Wound Healing

**DOI:** 10.3390/ijms24032936

**Published:** 2023-02-02

**Authors:** Haifeng Ye, Yuval Rinkevich

**Affiliations:** Institute of Regenerative Biology and Medicine, Helmholtz Zentrum München, Max-Lebsche-Platz 31, 81377 Munich, Germany

**Keywords:** fascia layer, biomaterials, new target, wound healing

## Abstract

As the first barrier of the human body, the skin has been of great concern for its wound healing and regeneration. The healing of large, refractory wounds is difficult to be repaired by cell proliferation at the wound edges and usually requires manual intervention for treatment. Therefore, therapeutic tools such as stem cells, biomaterials, and cytokines have been applied to the treatment of skin wounds. Skin microenvironment modulation is a key technology to promote wound repair and skin regeneration. In recent years, a series of novel bioactive materials that modulate the microenvironment and cell behavior have been developed, showing the ability to efficiently facilitate wound repair and skin attachment regeneration. Meanwhile, our lab found that the fascial layer has an indispensable role in wound healing and repair, and this review summarizes the research progress of related bioactive materials and their role in wound healing.

## 1. Introduction

The skin is the largest organ in the body and has multiple functions; its main function is to serve as a protective barrier. It protects the body from harmful chemicals, UV radiation, and pathogenic microorganisms. Among its diverse functions, the skin also synthesizes vitamin D, regulates body temperature, and prevents water loss. With over millions of burn victims, bedsores, venous stasis, and chronic skin ulcers such as diabetic foot reported each year in the United States and tens of millions of burn victims reported in China each year, skin wound healing has piqued the interest of researchers worldwide. The cost of skin wound treatment varies depending on the strategy, and the annual medical cost of chronic wound treatment in the United States is approximately $6 to $15 billion. Therefore, the development of some new biomaterials targeted for application in the right position of the skin to accelerate wound healing and achieve scar-free healing is a problem that the research community is trying to solve today.

## 2. The Structure of the Skin

The skin is composed of three layers: the upper epidermis, the dermis, and the hypodermis (also known as the subcutaneous tissue), each with its own unique structure and function. Refer to Figure 1.

### 2.1. Epidermis

The epidermis is the first layer of the skin. Most epidermal cells are keratin-forming cells, but skin also contains melanocytes, Merkel cells, Langerhans cells, and inflammatory cells [1]. The outermost layer of the epidermis, also known as the stratum corneum, is composed of 10 to 30 layers of polyhedral, non-nucleated keratinocytes that, when damaged, block most bacteria, viruses, and other external substances from entering the body [2,3]. Melanocytes are scattered in the basal layer of the epidermis and produce melanin, the number and distribution of which are major factors in the formation of skin color. There are also Langerhans cells in the epidermis, which are tissue-resident macrophages of the skin, which are part of the skin’s immune system. Langerhans cells not only recognize exogenous substances and become fully functional antigen-presenting cells, but also play a role in skin metaplasia. The epidermis resists infection by pathogens in the environment and regulates skin humidity [4,5].

### 2.2. Dermis

The dermis is the second layer of the skin, a thick layer of dense fibrous and elastic connective tissue that gives the skin its elasticity and strength. The dermis is divided into two layers; the superficial layer adjacent to the epidermis is called the papillary dermis, whereas the deeper, thicker region is called the reticular dermis [6]. Different dermal fibroblasts display distinct roles in skin development, homeostasis and wound healing [7]. The dermis is composed of three main types of cells: fibroblasts, macrophages, and mast cells [8,9]. In addition to these cells, the dermis is composed of matrix components such as collagen (which provides strength), elastin (which provides elasticity), and extracellular matrix, an extracellular gel-like substance composed mainly of glycosaminoglycans (most notably hyaluronic acid), proteoglycans, and glycoproteins [10]. The dermis houses nerve endings, sweat glands, sebaceous glands, hair follicles, and blood vessels, the abundance of which varies across different skin locations.

### 2.3. Subcutaneous Tissue

This is the lowermost tissue layer of the skin, which consists mainly of loose connective tissue and contains larger blood vessels and nerves than in the dermis [11,12]. The main cell types that inhabit the subcutaneous tissue are fibroblasts, adipocytes, and macrophages. It is the main site of fat storage in the body, helping to insulate against cold and provide a protective filling layer [13]. The subcutaneous tissue contains a large number of adipocytes within a connective tissue layer termed as fascia [14,15]. The structure of mouse skin and human skin is roughly the same. The main differences are that mouse subcutaneous tissue has a very thin muscle layer, whereas human skin does not. Refer to Figure 2.

## 3. The Basic Process of Skin Wound Healing and Its New Finding

Skin wound healing progresses by peri-wound resident cells, and can be divided into three key phases: inflammatory phase, proliferation phase, and scar formation and remodeling phase [17]. Refer to Figure 3.

### 3.1. Inflammatory Phase

The inflammatory phase begins after wound formation. First, cells in the injured tissue release vasoactive substances to cause vasoconstriction to prevent further bleeding, fibrinogen forms an insoluble fibrin network, platelets, lymphocytes, and granulocytes form a blood clot to close broken blood vessels and form a protective film at the wound to prevent further contamination by pathogens. Immune cells around the wound secrete various inflammatory mediators such as tumor necrosis factor (TNF), interleukin (IL), and several other cytokines and growth factors, which together with regulatory signals promote the progression of wound healing [18].

### 3.2. Proliferative Phase

During the proliferative phase, endothelial cells of the vessel wall break through the basement membrane and migrate to the peri-wound area, dividing to form vascular buds and interconnecting to form new vascular pathways, which further form vascular networks and capillary rings to form granulation tissue. The transition of wound healing from the inflammatory phase to the vascularized granulation phase is achieved with the joint participation of various growth factors and cytokines such as fibroblast growth factor (FGF) to promote angiogenesis and fibroblast division [19], epidermal growth factor (EGF) can promote angiogenesis, fibroblast migration and proliferation, and enhance collagen deposition, thus facilitating wound healing to grow new and intact skin [20].

### 3.3. Repair Phase

The third phase of wound healing involves remodeling of wounded tissue and the eventual growth of new skin after a long period of repair. The conversion of granulation tissue production to wound re-epithelialization marks the beginning of dermal tissue remodeling. During this process, the disorganized extracellular matrix (ECM), originally composed of type III collagen and elastin, is replaced by an ordered ECM composed of type I collagen and elastin fibers, remodeling the strength and elasticity of the dermis [21]. Platelet-derived growth factor (PDGF) signaling and its receptors are involved in hair follicle morphogenesis [22], and the formation of a new epidermis covering the wound under the combined action of various factors marks the completion of the wound healing process.

### 3.4. New Perspective of Skin Wound Healing

The traditional theory of wound healing suggests that the wound ECM is mainly derived from fibroblasts migrating to the wound bed, where they proliferate and differentiate into myofibroblasts, and then myofibroblasts produce collagen to deposit the ECM de novo [23,24,25,26]. The recent findings from our laboratory show that the fascial layer in the skin is the dominant factor in establishing wound ECM [27,28]. Prior to wounding the skin, the subcutaneous fascia was injected with NHS-488 dye to specifically label the ECM of the fascial layer. We then made 5-mm-sized full-skin wounds on the dorsum of mice. On the seventh day, from the samples we observed that most of the contents of the wound bed ECM came from the labelled fascia (Figure 4), suggesting that the fascia is a repository for wound healing. We further found that this influx of fascia ECM into the wound bed was mediated by En1-positive fibroblasts residing in the fascia [27], and that N-Cadherin could promote collective migration of fascia fibroblasts to carry ECM into the wounds and promote wound healing [29]. We further found that the gap junction α1 protein (Connexin43) is essential for large wound repair, and when inhibited, fascial ECM transport into wounds is reduced [28], leading to reduced scar formation and laying the foundation for clinical treatment of scarless healing. We have recently identified a central role for p120 in regulating fascial mobilization and wound repair, and gene silencing of p120 in fascial fibroblasts reduces the transfer of fascial cells and extracellular matrix to the wound and promotes wound healing [30]. In our latest study, a multipotent fibroblast progenitor expressing CD201 was identified in fascia. It accelerates wound healing by generating multiple specialized cell types, from pro-inflammatory fibroblasts to myofibroblasts, in a spatiotemporally coordinated sequence when the skin undergoes injury [31]. In summary, fascia tissue plays a non-negligible role in deep skin wound healing, which will provide a new target for clinical wound healing treatment.

## 4. The Role of Biomaterials in Skin Wound Healing and Regeneration

### 4.1. Types of Biomaterials

Biomedical materials are materials used to diagnose, treat, repair, or replace diseased tissues or organs of living organisms or enhance their functions. There are many kinds of trauma repair materials, among which there are hundreds of biological scaffold materials and more than 1800 different kinds of products. The existing scaffolds can be divided into three categories according to the materials used for their production: natural biological scaffolds, synthetic organic scaffolds, and inorganic scaffolds, each with its own advantages and disadvantages. Therefore, the form and function of prosthetic materials should also be diverse. The existing biological scaffolds are summarized as follows, see Table 1.

### 4.2. Mechanisms by Which Bioactive Materials Promote Wound Repair and Skin Regeneration

#### 4.2.1. Influence on Immune Cell Behavior

The process of wound repair is closely related to the inflammatory response because immune cells not only play the role of phagocytosis of pathogens and clearance of necrotic tissues [32], but also participate in the tissue repair process by secreting cytokines, growth factors, and matrix metalloproteinases (MMP). Modulation of wound immune cells and inflammatory responses can be achieved using bioactive materials. Some bioactive materials create a pro-regenerative inflammatory response microenvironment by inducing M2-type polarization in macrophages. For example, the nanocomposite dressing shortens the inflammatory phase and recruits macrophages to enhance angiogenesis and accelerate wound healing in normal and diabetic rats [33], whereas Casomorphin-containing keratin scaffolds stimulate macrophage infiltration and accelerate whole skin wound healing in diabetic mice [34]. In addition to the recruitment of macrophages, it also has a polarizing effect on them. Aloe/chitosan nanohydrogels modulate macrophage polarization in wound healing and the balance between M1 and M2 macrophages [35]. Silver nanoparticle-laden collagen/chitosan scaffolds promote wound healing by modulating fibroblast migration and macrophage activation [36]. The drug-laden hydrogel promotes wound closure by scavenging ROS and promoting macrophage polarization to M2 phenotypic macrophages [37]. The Col I/SCS hydrogel promotes M1 to M2 macrophage polarization and activates macrophage trans-differentiation to fibroblasts [38], it also balances pro- and anti-inflammatory cytokines to accelerate diabetic chronic wound healing. The starPEG-GAG hydrogel effectively clears inflammatory chemokines MCP-1 (monocyte chemotactic protein-1), IL-8 (interleukin-8), MIP-1α (macrophage inflammatory protein-1α), and MIP-1β (macrophage inflammatory protein-1β) to promote chronic wound healing [39].

These biomaterials can modulate neutrophils in addition to macrophage polarization. For example, chitosan scaffolds increased the number of neutrophils in aged wounds and accelerated wound closure in pressure ulcers of aged mice [40]. In situ delivery of hydrogels of fibronectin reduced the number of neutrophils, decreased the inflammatory response of wounds, and accelerated wound healing [41]. Chitosan-alginate membranes and chitosan-capped silver nanoparticles (Ch/AgNPs) can reduce the number of neutrophils and stimulate fibrous proliferation and collagen formation and accelerate wound healing [42].

Mast cell has a non-negligible role in all phases of wound healing. During the hemostatic phase, MC enhances the expression of factor XIIIa in dermal dendritic cells through the release of TNF-α [43,44] and is also able to block the conversion of fibrinogen to fibrinase [45], thus contributing to hemostasis and clot formation and to clot stabilization. During the inflammatory phase, keratinocytes recruit MCs to the site by secreting stem cell factor [46] (SCF) and releasing monocyte chemotactic protein 1 (MCP-1) from macrophages [47]. MCs in turn release inflammatory mediators, mainly histamine, VEGF, interleukin (IL)-6, and IL-8, which contribute to increased endothelial permeability and vasodilation [48] and promote inflammation cells (mainly monocytes and neutrophils) to migrate to sites of inflammation for injury [49,50]. The recruited monocytes are converted into efficient phagocytes for function by MCP-1 released from MCs [51]. During the proliferative phase, MCs are able to activate fibroblasts and keratin-forming cells [52,53]. MCs are activated during the proliferative phase by IL-4, vascular endothelial growth factor (VEGF) to stimulate fibroblast proliferation, and basic fibroblast growth factor (bFGF) to produce new extracellular matrix (ECM) to fill the wound [54,55]. During the remodeling phase, MC-derived mediators, including fibroblast growth factor 2, VEGF, platelet-derived growth factor (PDGF), TGF-β, nerve growth factor (NGF), IL 4 and IL-8, which contribute to the process of neoangiogenesis, fibrinogenesis, or re-epithelialization repair [56,57,58]. Implanted biomaterials all interact with the heterogeneous inflammatory environment present at the site of injury, and they can modulate immune cells, fibroblasts, endothelial cells, and keratinocytes, thereby promoting wound healing [59]. Thus, the modulation of MC by biomaterials can be used to improve wound healing and has gradually become a potential therapeutic strategy to accelerate wound healing.

#### 4.2.2. Modulation of Non-Immune Cell Behavior

Although immune cells are essential for wound healing, the role of non-immune cells cannot be ignored. The use of bioactive materials can modulate non-immune cells on the wound surface, which can achieve faster wound healing and smaller scar formation. During the process of wound repair, fibroblasts in deep wounds are overactive, proliferating and differentiating into myofibroblasts, synthesizing collagen and other matrix molecules in large numbers, and keratin-forming cells are also proliferating and differentiating, migrating toward the wound center, continuously covering the wound surface and eventually forming scar tissue [25]. Therefore, regulating the behavior of fibroblasts, inhibiting its differentiation to myofibroblasts, and accelerating the proliferation and migration of keratin-forming cells are important strategies to accelerate wound healing and reduce scar formation [60]. However, the application of drugs alone or the use of physical methods without the purpose of scar healing because the factors regulating the behavior of fibroblasts and keratin-forming cells are numerous and complex processes, involving not only the inflammatory response of immune cells, but also the structure, mechanical forces, and molecular composition of the microenvironment in which the cells are located. Bioactive materials offer significant advantages in modulating Fb and keratin-forming cell behavior.

Collagen-based dermal substitutes such as Integra^®^ and PELNAC^®^ that have been marketed can inhibit the formation of scars. They regulate the adhesion behavior of myofibroblasts through the GFOGER and GLOGER polypeptides in the scaffold, inhibit the aggregation of myofibroblasts, and reduce the contraction of the wound [61]. However, dermal substitutes such as Integra^®^ did not inhibit myofibroblast differentiation, whereas some decellularized matrix materials showed an impediment to myofibroblast differentiation in wounds [62,63]. The use of drug-loaded bioactive materials to modulate trabecular non-immune cell behavior is another way to promote regenerative trabecular repair. Scaffolding based on papaya seed mucus can promote fibroblast proliferation and differentiation of h-ASCs into keratin-forming cells, thus accelerating wound healing [64]. In addition, we can obtain better results by adding some active substances to the biomaterial. For example, the use of sulfonated polyether ether ketone (SPEEK) nanofiber scaffolds, modulating the delivery of aloe vera promotes the highest proliferation potential of HaCaT keratin-forming cells and fibroblasts to accelerate skin wound regeneration [65]. Three-dimensional chitosan dressing, accelerates the rate of skin wound healing by stimulating the migration, invasion and proliferation of relevant skin-resident cells [66]. Collagen 3D scaffolds containing Mesenchymal stromal cell-derived factors promote the healing phenotype by maintaining adhesion and proliferation of keratin-forming cells and fibroblast differentiation [67]. Engineered skin substitutes (ESS) consisting of autologous fibroblasts and keratin-forming cells attached to collagen-glycosaminoglycan (GAG) scaffolds are effective adjuncts for the treatment of large burns [68]. In addition, chitosan scaffolds, decellularized dermal scaffolds, and biodegradable microspherical polymer scaffolds co-cultured with keratin-forming cells and fibroblasts can promote wound healing [67,69,70,71]. Finally, some nanofibers can successfully direct cell migration and proliferation [72], and AgNPs can improve wound closure by promoting the proliferation and migration of keratin-forming cells and driving the differentiation of fibroblasts into myofibroblasts [73]. Collagen scaffolds loaded with zinc oxide-curcumin nanocomplexes significantly upregulated the activity of TGF-β3 expressed by rat burn wound cells and promoted scar-free wound healing [74].

#### 4.2.3. Promotion of Skin Attachment Regeneration

In addition, bioactive materials show high potential for application in skin attachment regenerative medicine techniques, which can complete skin regeneration by promoting the regeneration of hair follicles, sebaceous glands, sweat glands, and skin vessels and nerves.

Lee et al. found that raising epidermal and dermal cells from neonatal rats in integral dermal substitutes made from type 1 collagen resulted in visible hairs 11–15 days after transplantation [75]. Qi et al. developed a silk gel-based hydrogel for wound dressing for full skin injury repair and found that it recruited mesenchymal stem cells to the injury site for skin attachment regeneration [76]. Liu and colleagues found that citrate dressings have bioactive anti-inflammatory, antibacterial, and hemostatic effects, and can also regulate macrophage polarization, ultimately accelerating wound healing and hair follicle regeneration [77]. Some composites have been found to be effective for hair follicle regeneration. For example, chitosan/LiCl composite scaffolds promote skin regeneration in total skin wounds [78]. Curcumin nanoparticles incorporated into PVA/collagen composite membranes promote hair follicle regeneration and early epithelial re-formation for wound healing [79]. Zhang et al. developed a composite film releasing Qu, Cu, and Si ions with synergistic stimulation of hair follicle regeneration and promotion of hair follicle regeneration [80]. However, in in vitro experiments in which human-derived outer sheath keratin-forming cells and dermal papilla cells were grown within a matrix gel, we found that epidermal cyst-like cell spheres and tubular structures developed but could not form intact hair follicles [81]. Abaci et al. successfully regenerated skin tissue containing hair follicles by transplanting hair papilla cells into a type I collagen gel containing dermal fibroblasts through 3D bio-printing and formed microvasculature [82]. This technique laid a solid foundation for in vitro culture of hair follicles.

Li et al. found that matrigel promotes the differentiation of human epithelial cells of exocrine sweat gland tissue origin into exocrine sweat gland tissue [83]. Matrigel basement membrane matrix induced exocrine sweat cells to reconstruct sweat-like structures in nude mice [84,85,86], and Huang et al. used gelatin microspheres (containing epidermal growth factor [EGF]) as a multifunctional vehicle on which to culture sweat gland cells (SGC) and transplant SGC-microsphere complexes (SMC) onto whole skin wounds in a thymus-free mouse model and found that SMC could differentiate into sweat gland-like structures within a mixed matrix in vitro [87]. Chao et al. developed saccharose nanofibers (SCNF) with 3D gel structures to provide sweat gland and hair follicle growth in healing wounds, indicating that the wounds had healed as functional tissues [88]. Diao et al. used epithelial cells from sweat glands in the dermis of adult mouse paw pads to embed in matrix gel and form sweat gland-like organs (SGOs), which were transplanted into sweat gland-injured mouse back wounds and paw pads, respectively; they were found to have epidermal and sweat gland regeneration potential [89].

The development of different biological scaffolds to enhance vascular regeneration of wounds is also one of the effective methods to accelerate wound healing. Keratin-forming cells cultured on bladder matrix scaffolds can increase angiogenesis and aid in rapid wound healing [90]. Functionalizing collagen-based scaffolds with platelet-rich plasma were found to have the ability to promote angiogenesis in in vitro and in vivo assays [91]. Li et al. developed silica-based nanocomposite hydrogel scaffolds that promoted vascular regeneration by enhancing HIF-1α/VEGF expression in diabetic wounds, significantly accelerating wound healing and skin tissue regeneration [92]. Wang et al. also fabricated an injectable adhesive thermosensitive multifunctional polysaccharide dressing (FEP) that accelerated wound healing by stimulating the angiogenic process in wound tissue [93]. Nano-bioactive glass, a biomaterial used to induce angiogenesis in tissue engineering applications, has been highly sought after in recent years [94]. The addition of relevant microscale metal ions to nano-bioactive glasses can promote angiogenesis and accelerate wound healing [95]. For example, the addition of copper, cobalt, and cerium ions can promote endothelial cell proliferation, migration, stimulate HIF-1α, upregulate FGF and VEGF expression, and enhance the affinity of angiopoietin for endothelial cells [96,97,98,99,100]. 

## 5. Fascia Layer—A New Target for Biomaterials

The study of fasciology is not yet fully researched, and there is even a lot of potential to study. Fascia is a layer of fibrous connective tissue that runs throughout the body, lining the skin and encasing muscles, muscle groups, blood vessels, nerves, and internal organs [15]. There are three types of fascia called superficial, deep, and visceral fascia, which extend up and down the body [101]. The main body of fascia is fibrous connective tissue, which can contain fatty tissue, blood vessels, and nerves. Loose connective tissue is mainly found in the superficial and visceral fascia; dense connective tissue is mainly found in the deep fascia. The superficial fascia, also known as the subcutaneous fascia, is composed of loose connective tissue containing fat, superficial veins, dermal nerves, and superficial lymph nodes and lymphatic vessels. The superficial fascia is the lowermost layer of skin in almost all areas of the body and then runs through the subcutaneous adipose tissue, extending upward to mix with the reticular dermis [102]. The deep fascia, also known as the myo-fascia or fascia propria, is located deep within the superficial fascia, and is composed of dense connective tissue that is continuous throughout the body. The deep fascia is surrounded by muscles or muscle groups, glands, large blood vessels, and nerves to form a fascia sheath. The deep fascia of the extremities extends between the muscles and connects to the bone and is an important component of the body’s locomotor system [103]. The visceral fascia, also known as the sub-plasmatic fascia, suspends the organs in their cavities and encases them in a connective tissue membrane layer. Each organ is covered by a double layer of fascia; these layers are separated by a thin plasma membrane. Visceral fascia has an important role in the stability and movement of the viscera [104]. Visceral fat, also known as organ fat or intra-abdominal fat, is located in the peritoneal cavity and fills the space between the viscera and the trunk, as opposed to subcutaneous fat, which lies beneath the skin, and visceral fat, which is scattered throughout the abdominal cavity [105]. Like subcutaneous fat, visceral fat has a lot of fascia tissue running through it, and this visceral fascia tissue not only ensures. These visceral fascia tissues not only ensure that the viscera are not adherent to each other, but also run through the visceral fat to provide stability to the viscera.

As the fascia layer plays a crucial role in the process of skin wound repair, the contraction of the fascia leads to the closure of the wound. Moreover, the fascia layer is a layer of loose connective tissue under the skin, rich in blood vessels and nerves. Targeted regulation of fascia can intervene in wound healing and guide the clinical treatment of chronic wounds and diabetic wounds. At present, biomaterials mainly act on the epidermis and dermis, so it is a new direction to develop biomaterials to target the fascia layer.

## Figures and Tables

**Figure 1 ijms-24-02936-f001:**
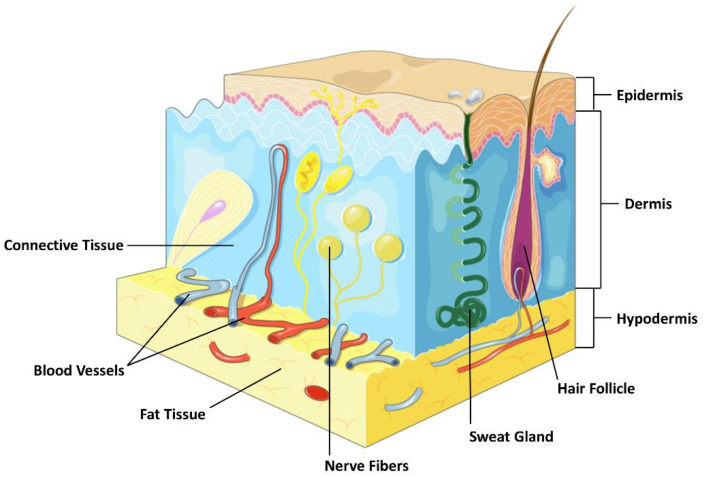
The structure of the skin.

**Figure 2 ijms-24-02936-f002:**
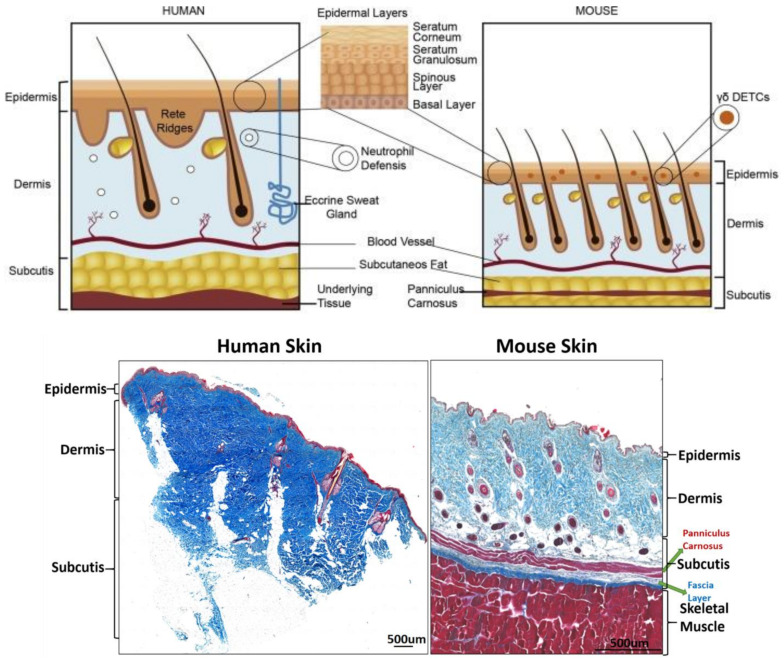
Histological differences between mouse and human skin. Above is a cartoon depiction of the difference between mouse and human skin. This image is a reference image [16]. Below is the histological difference between mouse and human skin, scale bar 500 μm.

**Figure 3 ijms-24-02936-f003:**
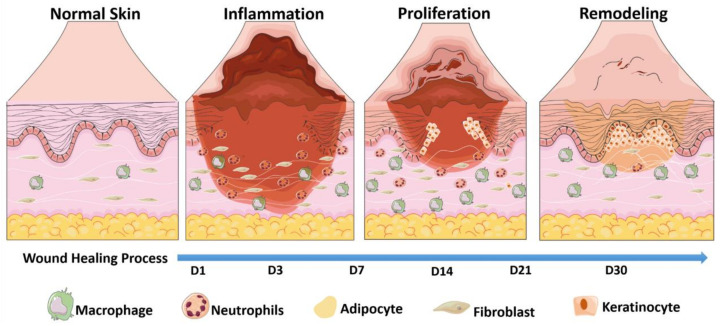
The key phases of wound healing.

**Figure 4 ijms-24-02936-f004:**
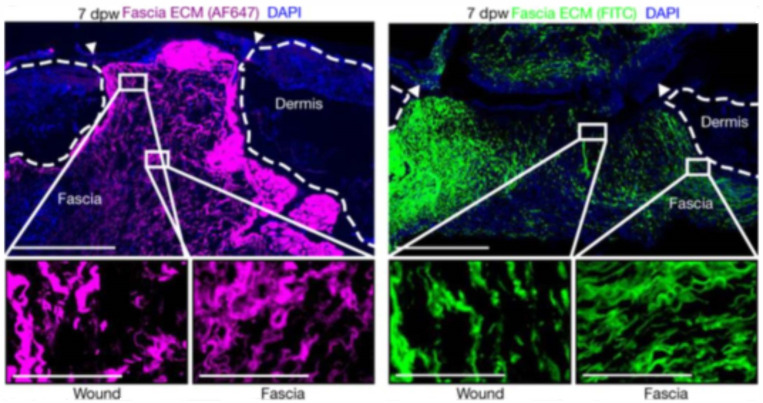
This image is a reference image [27], fluorescence images of wounds on day 7 of fascia labelling experiment, nuclei (blue), labelled fascia (magenta, green), scare bar: 500 µm (top), 100 µm (bottom).

**Table 1 ijms-24-02936-t001:** Advantages and disadvantages of different scaffolds used in skin tissue engineering.

Scaffold Types	Advantages	Disadvantages	Future Prospects
Porous scaffolds	High porosity provides a suitable environment for extracellular matrix (ECM) secretion and nutrient supplies to the cells. Pore sizes specific to the cell types prevent clustering of the cells, thus avoiding necrotic center formation.	Time consuming post-manufacturing cell inoculation, low cell viability, and high cost.	Improvement in the connectivity of pores and thereby the structure of the scaffolds is required.
Fibrous scaffolds	Highly microporous structure is best suitable for cell adhesion, proliferation, and differentiation. Low inflammatory response upon implantation.	Surface functionalization is required to create the nanofibers of these scaffolds.	Drugs and biological molecules such as proteins, genes, growth factors, etc., can be incorporated in fibrous scaffolds for release applications.
Hydrogel scaffolds	Highly biocompatible and controlled biodegradation rate.	Limited mechanical strength due to soft structures.	Degradation behavior of the hydrogels and tenability should be well-defined. Hydrogels incorporating growth factors to facilitate cell differentiation.
Microsphere scaffolds	Easily fabricated with controlled physical characteristics suitable for slow or fast drug delivery. Provides enhanced cell attachment and migration properties.	Microsphere sintering methods are sometimes not compatible to the cells and reduces the cell viability.	These scaffolds can be used as a target specific delivery vehicle for the drugs such as antibiotics, anti-cancer, etc.
Composite scaffolds	Highly biodegradable and offer mechanical strength. Greater absorbability.	Acidic byproducts are generated upon degradation. Poor cell affinity. Require tedious efforts to develop composite scaffolds.	Nano-bioceramic and polymer composites with faster degradation are currently being developed.
Acellular scaffolds	Native ECM is retained, and thus normal anatomical features are maintained. Less inflammatory and immune response with higher mechanical strength.	Incomplete decellularization is required to avoid immune responses.	Such scaffolds hold promise towards developing artificial organs.
Extracellular Matrix-Based Scaffolds	Retains native ECM, less inflammation and immune response.	Rapid degradation and poor mechanical properties.	Scaffold cross-linking, blending with other biomaterials, and the addition of bioactive substances to create multifunctional scaffolds for ECM.
Porous Scaffolds	Porous scaffolds have excellent load-carrying capacity, many of which can payload cargos with relatively large sizes.	Time consuming post-manufacturing cell inoculation, low cell viability, and high cost.	There is a need to improve the connectivity of the pores and the structure of the scaffolds.
Fibrous Scaffolds	Suitable for cell adhesion, proliferation, and differentiation with low inflammatory response, giving the cells their typical in vivo morphology.	Surface functionalization required.	Hybrid fiber scaffolds with enhanced properties (biomechanical, physico-chemical and biological) need to be developed.
Microsphere Scaffolds	With spatial expansion, temporal duration control and site targeting.	Reduces cell viability and has certain incompatibility with cells.	Development of an efficient drug delivery system.
Hydrogel Scaffolds	Hydrogel scaffolds are highly hydrophilic, flexible, biocompatible, and biodegradable.	Limited mechanical properties, difficult purification and sometimes pathogenic transmission and immunogenicity.	Development of a “smart” 4D hydrogel.
Nanoparticle scaffolds	Nanoparticle scaffolds have high mechanical properties, antibacterial ability.	Nanoparticle scaffolds may be toxic, cancerous, and teratogenic.	More efficient and safe nanoparticle scaffolds to be developed.
Polymer Scaffolds	Better biocompatibility, reproducible mechanical properties, processability, and low price.	Sometimes triggers an immune response and toxicity and poor cell affinity. Require tedious efforts to develop composite scaffolds.	Faster and safer polymer scaffolds are currently being developed.

## Data Availability

Not applicable.

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
