# Peer review of "Fascia Layer—A Novel Target for the Application of Biomaterials in Skin Wound Healing"

_ijms, 2023, doi:10.3390/ijms24032936_

Round 1

Reviewer 1 Report

The paper seems to be a repetition of other papers related to skin. no novelty seems to be added to the paper

authors can remodel the paper in a more innovative way. 

Add more delivery systems

Author Response

Point 1: The paper seems to be a repetition of other papers related to skin. no novelty seems to be added to the paper

Response 1: Thank you very much for your careful reading and kind reminder, in this review we would like to propose that the fascia layer will be a new target of biomaterials in skin healing. Although fascia has been well known throughout the body, its crucial role in skin wound healing has only been known in recent years, and there is still very little research on the role of fascia in wound healing. In recent years, the research on biomaterials in skin healing has been very hot, but most of them are aimed at the epidermis and dermis of the skin, and there is almost no research on biomaterials targeting the fascia layer, so the purpose of this review It is proposed that the fascia layer will be a new target for biomaterials in wound healing.

Point 2:  authors can remodel the paper in a more innovative way. Add more delivery systems

Response 2: Thank you very much for your suggestion, we have properly refined and concentrated the structure and function part of the front skin. Then I added a part of the introduction of fascia and the effect of biomaterials on mast cells in wound repair. As for your suggestion to add a delivery system, I did not add it in this review, because in this review we mainly talk about bioscaffold materials, if the delivery system is added, it will be very complicated. We will address the role of delivery systems in skin healing separately in the next review.

Reviewer 2 Report

The review is very interesting. Finally authors recognize in the microenvironment a target for wound healing. The review is easy to read and it's well wriiten. Since the authors speak aboute the role of cellular infiltrate and the response of this during the different phases of wound healing it would seem appropriate for the authors to also talk about other cell types involved as well as the evaluation of the responses of macrophages, fibroblasts and endothelial cells. In this regard, a central role is played by mast cells, which play an important role in all phases of wound healing. This idea should also be considered in section 4.2 of this review. Although the description of the skin should be a less elaborate section considering the fact that the structure and functions are widely known

Author Response

Point 1:  In this regard, a central role is played by mast cells, which play an important role in all phases of wound healing. This idea should also be considered in section 4.2 of this review. 

Response 1: Thank you very much for your careful reading and kind reminder, as mast cells play such an important role in all stages of wound healing, we added this section in section 4.2.1 from line 204 to line 235.

Point 2:  Although the description of the skin should be a less elaborate section considering the fact that the structure and functions are widely known

Response 2: Thank you very much for your suggestion, we have refined and condensed this section properly.

Reviewer 3 Report

It is a well written manuscript with appriopriate background. In my opinion the manuscript is ready for submission after some minor changes. For example the brackets with references are sometimes different and sometimes with or without distance, before or after full stop. 

I was wondering if authors could describe the connection between fascia and visceral fat?

Author Response

Point 1: It is a well written manuscript with appriopriate background. In my opinion the manuscript is ready for submission after some minor changes. For example the brackets with references are sometimes different and sometimes with or without distance, before or after full stop. 

 Response 1: Thank you very much for your careful reading and friendly reminder. We have standardized the formatting of all parentheses with references.

Point 2: I was wondering if authors could describe the connection between fascia and visceral fat?

Response 2: Thank you very much for your suggestion, we added some introduction of fascia and the connection between fascia and visceral fat in section 5 from line 377 to line 414.

Round 2

Reviewer 1 Report

Comments have been taken into consideration and can be accepted now